# Ultrastructure of human brain tissue vitrified from autopsy revealed by cryo-ET with cryo-plasma FIB milling

Benjamin C. Creekmore [1,2,3,5], Kathryn Kixmoeller[2,3,5], Ben E. Black [2,4], Edward B. Lee [1] ✉ & Yi-Wei Chang [2,4] ✉

Ultrastructure of human brain tissue has traditionally been examined using electron microscopy (EM) following fixation, staining, and sectioning, which limit resolution and introduce artifacts. Alternatively, cryo-electron tomography (cryo-ET) allows higher resolution imaging of unfixed cellular samples while preserving architecture, but it requires samples to be vitreous and thin enough for transmission EM. Due to these requirements, cryo-ET has yet to be employed to investigate unfixed, never previously frozen human brain tissue. Here we present a method for generating lamellae in human brain tissue obtained at time of autopsy that can be imaged via cryo-ET. We vitrify the tissue via plunge-freezing and use xenon plasma focused ion beam (FIB) milling to generate lamellae directly on-grid at variable depth inside the tissue. Lamellae generated in Alzheimer's disease brain tissue reveal intact subcellular structures including components of autophagy and potential pathologic tau fibrils. Furthermore, we reveal intact compact myelin and functional cytoplasmic expansions. These images indicate that plasma FIB milling with cryo-ET may be used to elucidate nanoscale structures within the human brain.

Transmission electron microscopy (transmission EM) has long been used to inspect cellular structures that are not readily distinguishable via light microscopy. While allowing relatively high-resolution imaging, this technique historically applies chemical fixation, dehydration, and heavy element staining to ensure adequate visualization of structures. However, these methods can perturb native cellular architecture, especially delicate structures such as cytoplasmic channels within myelin[1,2]. High-pressure freezing (HPF) followed by freeze substitution improves sample preservation, but these approaches still involve organic solvent incubation, resin embedding, and mechanical sectioning samples for EM visualization, which reduces attainable resolution[3–6].

Cryogenic transmission electron microscopy (cryo-EM) eliminates the requirement of fixation or staining, allowing for the visualization of details not attainable by traditional EM methods. In the field of neurodegeneration, cryo-EM has mostly been performed on extracted protein samples[7–10]. However, these aggregates have all been removed from their native environment and thus do not provide a complete understanding of the associated cellular processes. For phenomena difficult to isolate or recapitulate in vitro, such as vesicular architecture and myelination, traditional EM methods remain the most informative to date.

Cryo-electron tomography (cryo-ET) has emerged as a technique for using cryo-EM to image cells and tissues to better understand the

[1]Translational Neuropathology Research Laboratory, Department of Pathology and Laboratory Medicine, Perelman School of Medicine, University of Pennsylvania, Philadelphia, PA, USA. [2]Department of Biochemistry and Biophysics, Perelman School of Medicine, University of Pennsylvania, Philadelphia, PA, USA. [3]Biochemistry and Molecular Biophysics Graduate Group, Perelman School of Medicine, University of Pennsylvania, Philadelphia, PA, USA. [4]Institute of Structural Biology, Perelman School of Medicine, University of Pennsylvania, Philadelphia, PA, USA. [5]These authors contributed equally: Benjamin C. Creekmore, Kathryn Kixmoeller. ✉e-mail: edward.lee@pennmedicine.upenn.edu; ywc@pennmedicine.upenn.edu

                                                                                                                          

context and 3D organization of structures. In cryo-ET, tilt series are collected at regions of interest by taking images at different angles and then reconstructing these images into 3D tomograms. Importantly, to attain high resolution, cryo-ET, and cryo-EM in general, rely on adequate vitrification to avoid perturbation of structures by crystalline ice and require samples thin enough to allow electrons through the sample. For purified aggregates or thin cells in culture, vitrification is relatively easily achievable by rapid plunge-freezing in liquid ethane directly on a cryo-EM grid. However, plunge-freezing alone is only effective to less than about 10 μm thickness[11]. HPF can vitrify thicker samples, such as tissues[5], however, it is low throughput and makes sample thinning prior to cryo-ET challenging. Most approaches using HPF for cryo-ET employ either cryo-sectioning which leaves surface artifacts and distortions[12], or cryo-liftout, which is low throughput and technically challenging[13]. To avoid HPF-associated challenges, it was recently shown that the addition of glycerol can vitrify some samples by plunge-freezing to about 200 μm thick[14], but this seems to be sample-dependent (described below).

Once vitrified, cryo-focused ion beam (cryo-FIB) milling is currently the method of choice for thinning cryo-EM samples because it introduces fewer artifacts compared to cryo-sectioning[12,15]. Gallium is the ion source most commonly used for cryo-FIB milling of biological samples. However, gallium lacks the power to efficiently remove large amounts of material and, over time can alter sample architecture due to the depth of gallium interaction with samples[16]. In contrast, xenon plasma-based cryo-FIB milling, primarily used in material science, is capable of efficient thinning of large samples and has reduced sample interaction depth compared to gallium[17–19].

Motivated by the aforementioned technical challenges and the fact that conventional tissue preservation methods at the time of autopsy (chemical fixation or −80 °C storage) introduce artifacts or damage, we sought to develop an approach that permits cryo-ET imaging of human brain tissue obtained at the time of autopsy without prior fixation or freezing. We describe a protocol for vitreous plunge-freezing of human brain tissue at least 180 μm thick. We further present a method for using xenon plasma-based cryo-FIB milling at currents never previously used in a biological context to generate lamellae suitable for cryo-ET imaging. These lamellae are generated directly on cryo-EM grids at variable depths into the tissue and are not limited to starting at the tissue surface. We demonstrate the capability of this approach by providing examples of the visualization of native subcellular compartments, potential disease-associated tau aggregates, and the intricate myelin architecture—including insights into the structural organization of myelin basic protein.

## Results
### Acquisition and vitrification of human brain tissue
In order to image human brain tissue in a near-native condition in the absence of freezing or fixation artifacts, we obtained unfixed, never previously frozen samples from the middle frontal cortex of several individuals at the time of autopsy (box in Fig. 1a, b). Gray matter (box in Fig. 1b) was further cut into small pieces <3 mm in width to approximate the dimensions of a cryo-EM grid. Tissue was embedded in cool low-melting point agarose (Fig. 1c), and then vibratome sectioned to a final thickness of about 100 μm (Fig. 1d). Due to the malleable and soft consistency of fresh brain tissue, consistently sectioning thinner than roughly 100 μm was difficult. As measured by scanning EM and FIB imaging in the subsequent FIB milling step, tissue sections prepared this way ranged from 30 to 280 μm thick.

We sought to determine whether unfixed human brain tissue could be vitrified via plunge-freezing (Fig. 1e) since this approach would allow for high-throughput and simpler on-grid lamella generation as opposed to the technically challenging cryo-liftout[20], ultimately making this approach more widely accessible. We attempted to vitrify tissue by plunge-freezing after incubation in 10% glycerol for 15 min

following recent protocols[14]. However, lamellae in these samples were not vitreous, and images contained Bragg reflections caused by electron diffraction due to crystalline ice (pink arrows Supplementary Fig. 1a). We next increased the glycerol to 20% and added 1 M trehalose based on the ability of trehalose to augment glycerol cryoprotection of stem cells[21,22]. We successfully used this method with four separate autopsy cases with incubation times down to 70 min for samples 180 μm thick. Using this protocol, we did not see evidence of crystalline ice within 17 lamellae imaged by cryo-EM that had an average $xy$ area of approximately 1,900 μm$^2$ per lamella (Supplementary Fig. 1b, Supplementary Fig. 2). We do not rule out that shorter incubations in cryoprotectant or thicker samples may also allow full vitrification.

### On-grid xenon plasma FIB milling of vitrified human brain tissue
To date, a significant obstacle in FIB milling of large samples has been the substantial amount of time necessary to remove large areas of material. Biological samples have almost exclusively been milled on gallium-based FIBs, but these instruments have limited milling rates and consequently are not ideal for large tissue samples. We therefore established a Tescan S8000X plasma FIB (PFIB)/scanning electron microscope (SEM) with Leica VCT and VCM for cryo-FIB milling using xenon plasma. FIB current is the rate at which ions are projected at a sample, and current is generally proportional to the milling rate for a given ion source. Xenon PFIB can achieve higher currents and mills faster at a given current than gallium FIB for most materials[23]. Cryo-PFIB has only recently been used for biological samples and in those cases, used only beam currents also accessible via gallium FIB[17].

In addition to the obstacles posed by FIB milling rates, large samples, especially those that are thick and cover most of an EM grid surface, pose a geometrical challenge for lamella generation and placement. Considerations for lamella placement include stability of the site and avoiding areas of potential mechanical damage from vibratome sectioning. It must also be feasible to remove all material from above and below the final lamella position, as the remaining material will impede subsequent cryo-EM imaging. Even material in front of or behind a lamella can impede imaging when the sample is tilted during tilt series acquisition. Recent methods have allowed for on-grid lamella generation in relatively large samples[14,24]. However, like most previous methods, they primarily generate lamellae at the surface of the specimen and so the depth into a specimen is determined by the length and angle of the lamella or a prominent sample edge. For cells and organisms such as *Caenorhabditis elegans* that can be vitrified intact[24], starting a lamella at the surface has minimal disadvantages. However, bulk tissue samples, like a human brain, require pre-processing with various blades to cut the tissue prior to vitrification. Therefore, we sought to develop a protocol allowing lamellae to be initiated from the interior of the tissue, thereby avoiding tissue disruption and artifacts present at pre-cut surfaces and allowing more flexibility in lamella placement which would be beneficial for correlation with light microscopy in the future.

First, the entire surface of the tissue was coated with a protective layer of organometallic platinum from the PFIB/SEM's gas injection system (Fig. 1f). Next, large amounts of tissue were removed to expose an internal tissue face (Fig. 1f, g) where lamellae can be generated at variable depth. This initial 'trench milling' was performed with the FIB oriented roughly orthogonal to the plane of the grid using a beam current of 1 μA, which is not achievable on currently available gallium FIB systems and, even if developed, would likely have a beam diameter too wide to be used for this purpose[25]. When positioned for near-orthogonal trench milling, the Gaussian tails of the 1 μA beam were about 80 μm measured from the edge of a milled area to where there is visible perturbation of the platinum coating. To prevent damage to the final lamellae, the beam current was lowered to 100 nA, about 80 μm from the desired lamella position. The 100 nA beam current, while available on gallium FIB systems, has a reduced probe size when using

                                                                                               

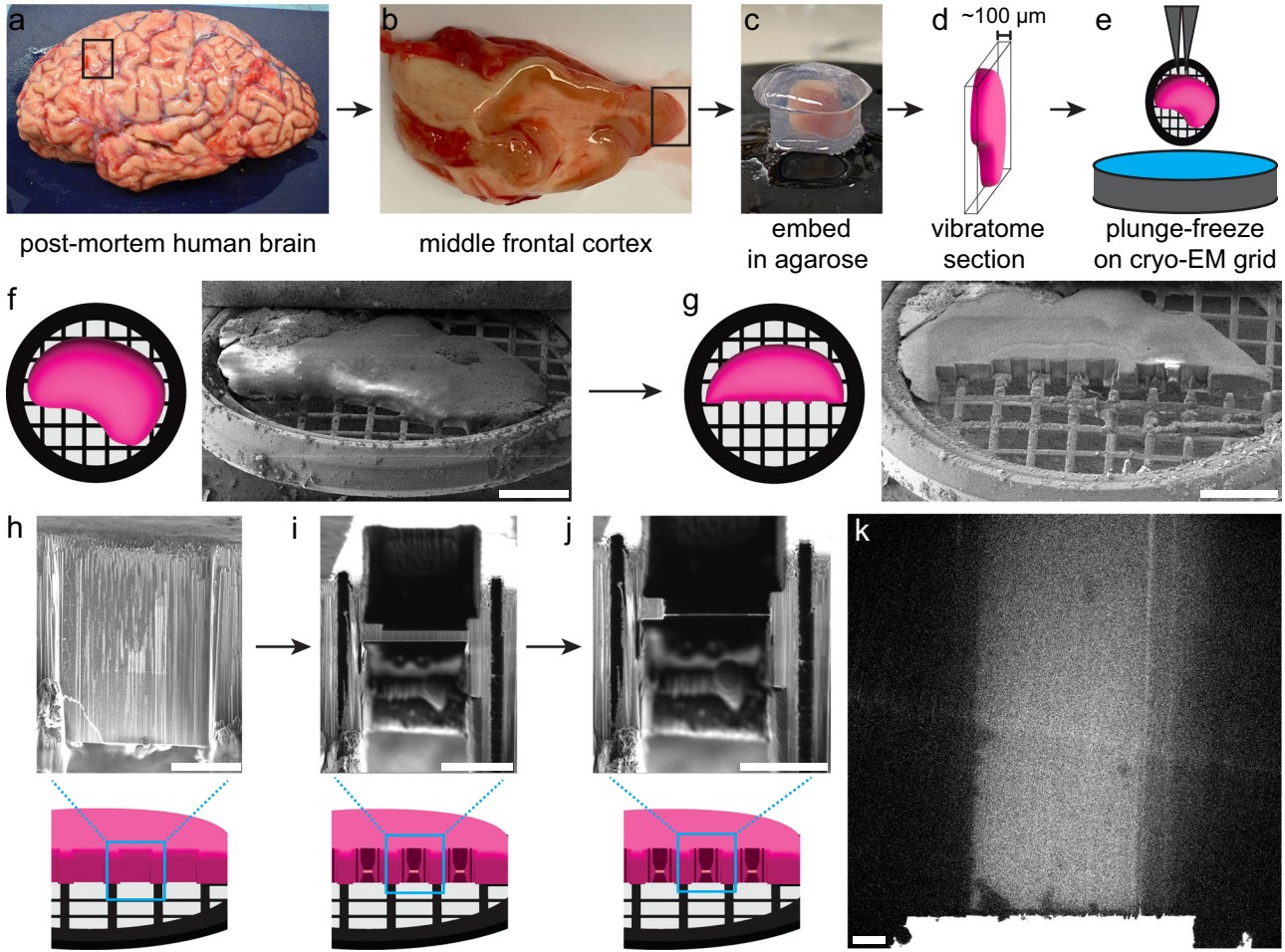

**Fig. 1 | Vitrification and cryo-FIB milling of unfixed, never previously frozen human brain tissue. a–e** Schematic of the procession of post-mortem brain tissue from gross sample (**a**) used for acquisition of middle frontal cortex (black box in **a**, **b**) to the isolation of cortical gray matter (black box in **b**) to embedding tissue in agarose molds (**c**) prior to vibratome sectioning (**d**) and incubation in cryoprotectant prior to plunge-freezing sample (**e**). **f**, **g** Schematic and example SEM image of tissue within FIB/SEM prior to milling (**f**) and after trench milling (**g**) by cryo-FIB. Scale bars, 500 μm. **h–j** Schematic and example FIB images of on-grid lamella milling to generate final lamellae (**j**). Scale bars, 50 μm. **k** Example transmission EM image of lamella suitable for cryo-ET. Scale bar, 5 μm. This protocol was used on four separate autopsy cases generating 17 different lamellae imaged by transmission EM.

xenon PFIB[25] and thus has a more precise interaction area. The 100 nA beam was also used to remove grid bars near the internal tissue face in the near-orthogonal milling position to prevent the bars from occluding imaging of final lamellae. To improve the milling rate of grid bars, the FIB beam was focused on the grid bar prior to this step. For the final step of trench milling, a 10 nA beam is used with a directional polishing pattern to smooth the front of the internal tissue face to prepare this newly exposed surface for platinum coating and lamella generation (Fig. 1g). Trench milling took on average about 7 hours but ranged from 5.5 to 11 h depending on the sample. The trench milling time was most dependent on the number of grid bars removed rather than sample thickness.

Once the trench milling step was completed, the sample was rotated to point the gas injection system needle directly at the newly exposed internal tissue face. The tissue face was iteratively coated with organometallic platinum. After each coat, the deposited platinum layer was cured by exposure to the 10 nA FIB—a practice common in material science cryo-FIB milling[26]. We empirically found that intermediate and iterative curing of deposited platinum compacted the platinum to prevent porous platinum deposition. In contrast, during early protocol development, platinum was deposited all at one time and cured at the end, similar to many described protocols for biological materials[14,24,27]. This led to porous platinum, which did not adequately protect the

sample, prevent curtaining, or prevent sample charging (Supplementary Fig. 3a, b). These issues prevented reliable final lamella polishing and led to visible radiation-like damage of lamellae that significantly reduced the tolerated electron dosage in the downstream transmission EM (Supplementary Fig. 3a). With the iterative approach to platinum deposition, we did not see damage to lamellae (Supplementary Fig. 3c). Additionally, iterative curing of platinum allowed initial platinum layers to better fill-in residual curtains on the exposed internal tissue face left by the 10 nA step of trench milling (Supplementary Fig. 3d).

After platinum coating, lamellae were milled into the tissue face at positions away from the vibratome-cut surfaces and preferentially targeted areas of smooth platinum. These lamellae were milled using an approach similar to other on-grid lamella generation methods, starting at 10 nA beam current and gradually reducing to 50 pA for final polishing (Fig. 1h–j)[14,17,24]. Final polishing was done for each lamella on a grid just prior to unloading. Each lamella took about 2–4 hours to mill, including time for final polishing. Expansion joints about 8 μm wide were also milled on both sides of the lamellae. Expansion joints serve to release mechanical stress due to slight movements or fluctuations in temperature and thus help preserve lamellae during sample transfer steps (Fig. 1i, j). Lamellae were milled to a final width of about 30–40 μm and a length of about 25 μm to more than 70 μm (Fig. 1j, k). We noticed a reduction in successful lamella completion and survival

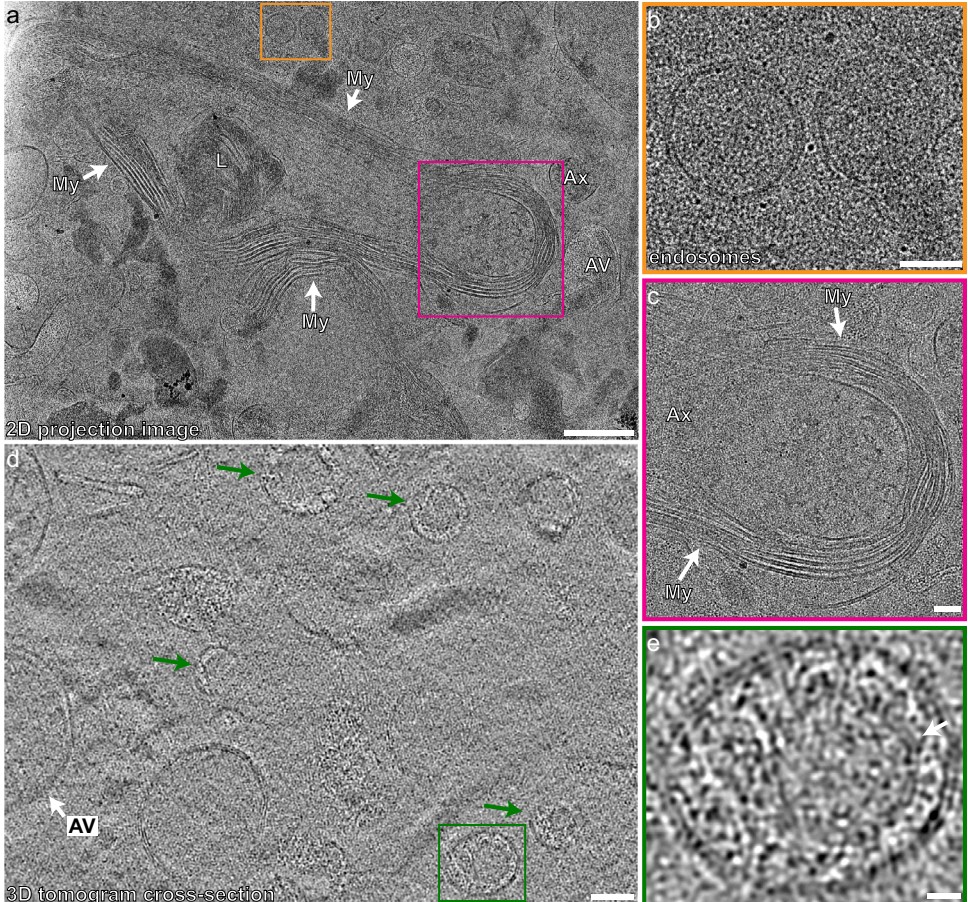

**Fig. 2 | Architecture of cellular and subcellular features is preserved within lamellae. a** 2D projection image at ×19,500 of an example lamella that shows the preserved architecture of a lysosome (L), unmyelinated axon (Ax), autophagic vesicle (AV), and myelin (My). Scale bar, 500 nm. The orange box expanded in **b** shows the preserved membrane architecture of two putative endosomes. The pink box expanded in **c** shows preserved myelin architecture around an axon. Scale bars, 100 nm for **b** and **c**. **d** cross-section of a denoised tomogram showing an autophagic vesicle (AV) and granular vesicles (green arrows). Scale bar 100 nm. The green box expanded in **e** shows a closer view of a granular vesicle with a potential inner membrane (arrow). Scale bar, 20 nm.

through sample transfer steps when lamellae were wider than 50 μm. With the final protocol, we attempted 19 lamellae, of which 5 did not survive milling, 5 did not survive the transfer and loading to cryo-EM, and 9 were imaged by cryo-EM.

While developing this approach, we attempted to use a 100 nA beam current for the initial step of lamella milling for increased efficiency. However, using the 100 nA beam current in the low-angle milling position resulted in an increased charging effect compared to starting with the 10 nA beam. Even with the improved iterative platinum deposition method, charging from using the 100 nA beam still prevented reliable lamella milling. The 100 nA beam at a low angle was less predictable than the 10 nA beam and often showed distortion of the milling pattern, likely due to charging effects or interaction of the Gaussian tails of the beam with the metal clip ring around the grid. While theoretically, the 100 nA beam can increase milling efficiency, the unpredictable nature of the beam in this context decreased the success rate and overall efficiency. We therefore opted to exclusively use beam currents 10 nA or lower throughout the low-angle lamella milling steps.

**Cryo-transmission EM of generated lamellae**
After milling, grids were transferred to a Titan Krios cryo-electron microscope for imaging. We found that lamellae prepared by the above method could survive the transfer from the PFIB to Titan Krios and were generally thin enough for cryo-EM imaging (Fig. 1k). Lamellae on which we acquired successful tilt series were on average 350 nm but

ranged from about 150–450 nm. Forty successful tomograms were reconstructed with the final protocol. 2D projection images of lamellae at various magnifications (Fig. 1k, Fig. 2a–c, Supplementary Fig. 1b, and Supplementary Fig. 4) and tilt angles (Supplementary Fig. 2i–p) showed no evidence of crystalline ice or gross damage within the lamella and revealed the intact cellular and sub-cellular milieu of unfixed, cryo-preserved human brain tissue (Fig. 2a). Importantly, many delicate structures often disrupted by conventional autopsy sample freezing/fixation or by mechanical cutting were found to be intact and well-preserved in our lamellae. For example, myelin and lipid membranes showed no signs of fracture, and vesicles maintained their rounded shape (Fig. 2b, c). From 2D images, subcellular structures could be identified, including putative endosomes, autophagic vesicles, and lysosomes (Fig. 2a–c and Supplementary Fig. 4a, b). We also visualized the repeating unit of myelin (two lipid bilayer membranes that appear as three distinct lines) preserved in our images (Fig. 2a, c and Supplementary Fig. 4). This has previously only been described in fixed and stained tissue[28].

To gain 3D insights, tilt series were acquired on lamellae. Tilt series were reconstructed into tomograms to reveal the 3D organization of cellular and subcellular structures (Fig. 2d, e, Figs. 3–4, and Supplementary Movies 1–5). The 3D data of tomograms also provided better visualization of subcellular membranous and proteinaceous structures. We describe some of the most prominent features from our tomograms and analyze the lipid bilayer of myelin to highlight examples of what can be studied by this method.

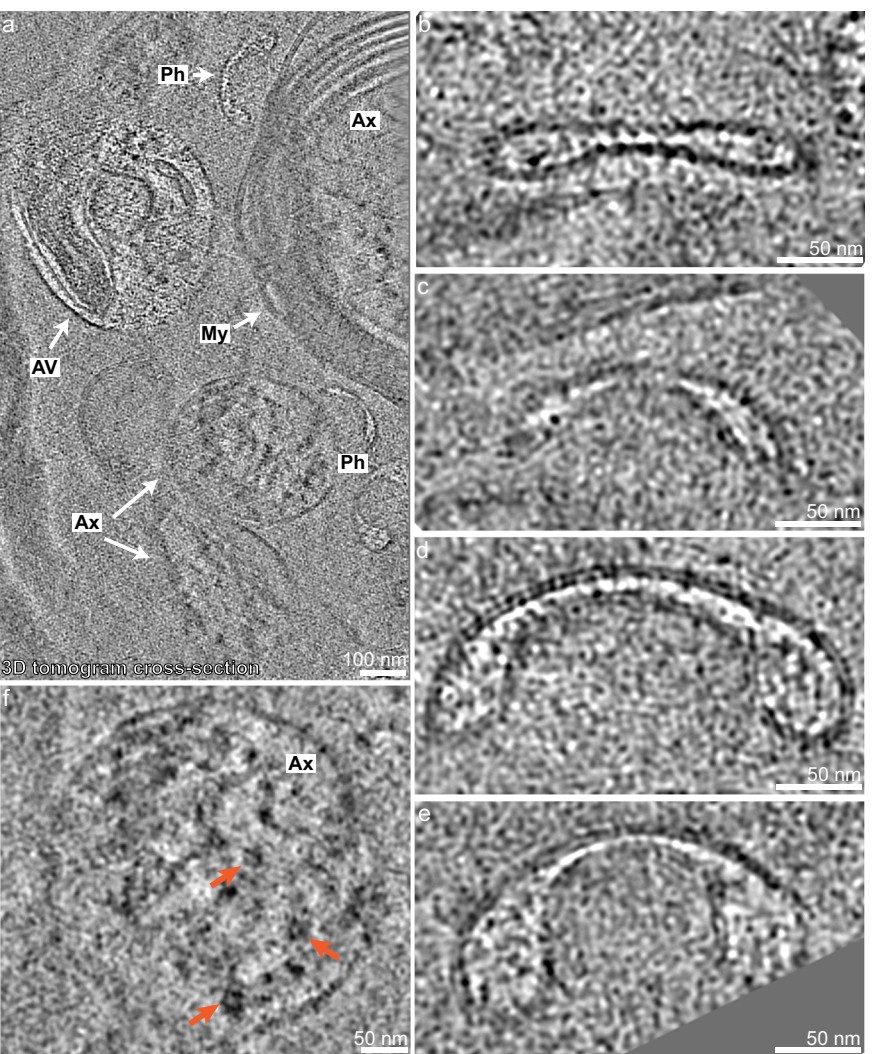

**Fig. 3 | Various subcellular features can be identified in lamellae within human brain tissue. a** Cross-section of a denoised tomogram showing an autophagic vesicle (AV) containing various cellular structures, phagophore-like structures (Ph), two unmyelinated axons (Ax), and one myelinated (My) axon (Ax). Scale bar, 100 nm. **b–e** Phagophore-like structures from the tomogram in **a** that show single-membrane structures with narrow centers and dilated ends. Scale bar, 50 nm. **f** Filamentous protein densities from an unmyelinated axon in **a** (orange arrow). Scale bar, 50 nm.

Membranous vesicles were abundant, including relatively large single-membrane bound vesicles containing heterogeneous cellular elements that may be autophagic vesicles, autophagolysosomes, or lysosomes, an example of which is shown in Fig. 3a (AV with white arrow). Granular vesicles of various sizes were visualized and are composed of an outer membrane surrounding a relatively electron-dense spherical core (Fig. 2d, e). In some examples, we observe a tentative, additional layer of membrane surrounding the dense core (Fig. 2e, arrow). Similar granular vesicles of the central nervous system reported by traditional EM are thought to contain neuropeptide[29], though the exact identity is unknown. In addition, we found crescent-shaped cross-sections of single-membrane structures, which are narrow in the center and dilated at the ends (Fig. 3a–e and Supplementary Movie 2). We observe different forms of these structures that we hypothesize to be different stages of biogenesis (Fig. 3b–e). The peripheral dilation becomes more pronounced as the structure expands and curves inward (Fig. 3b–e). This process seems to mimic phagophore biogenesis observed in yeast and cultured human cells, where there is a dilation of the rim of the forming phagophore[30,31]. These similarities suggest that the crescent-shaped features are phagophore-like structures. Future work may be able to shed light on the biogenesis and function of subcellular compartments directly in human tissue.

Beyond intracellular membranous structures, both myelinated and unmyelinated axons were visible in several tomograms. Axon cross-sections revealed filamentous densities parallel to the length of the axons (Fig. 3f, orange arrows). Additionally, inside a longitudinal cross-section of a myelinated axon, we observed long twisting filamentous structures that measured between 8 and 18 nm in width along the same filament (Fig. 4b–f and Supplementary Movie 4). The donor brain used for these tomograms (Fig. 1a) had a high level of Alzheimer's disease neuropathologic change, including a severe burden of tau and amyloid-β with TMEM106B present in the absence of TDP-43 or α-synuclein aggregates in the middle frontal cortex (Supplementary Fig. 5). Based on the dimensions of the filaments (Fig. 4d), the measured fibril crossover distance (Supplementary Fig. 6), the location and orientations within axons, our knowledge that tau was present at a high burden in the region sampled, and comparison to recently published images of tau fibrils in previously frozen brain tissue[32] and isolated extracellular vesicles (Supplementary Fig. 7)[33], these twisting filaments were most consistent with tau aggregates. We attempted subtomogram averaging of the filaments but were unable to obtain a reliable average due to the low number of filaments and few cross-sectional views, which were noted to allow for subtomogram averaging in other samples[32]. Nevertheless, this method shows the possibility of

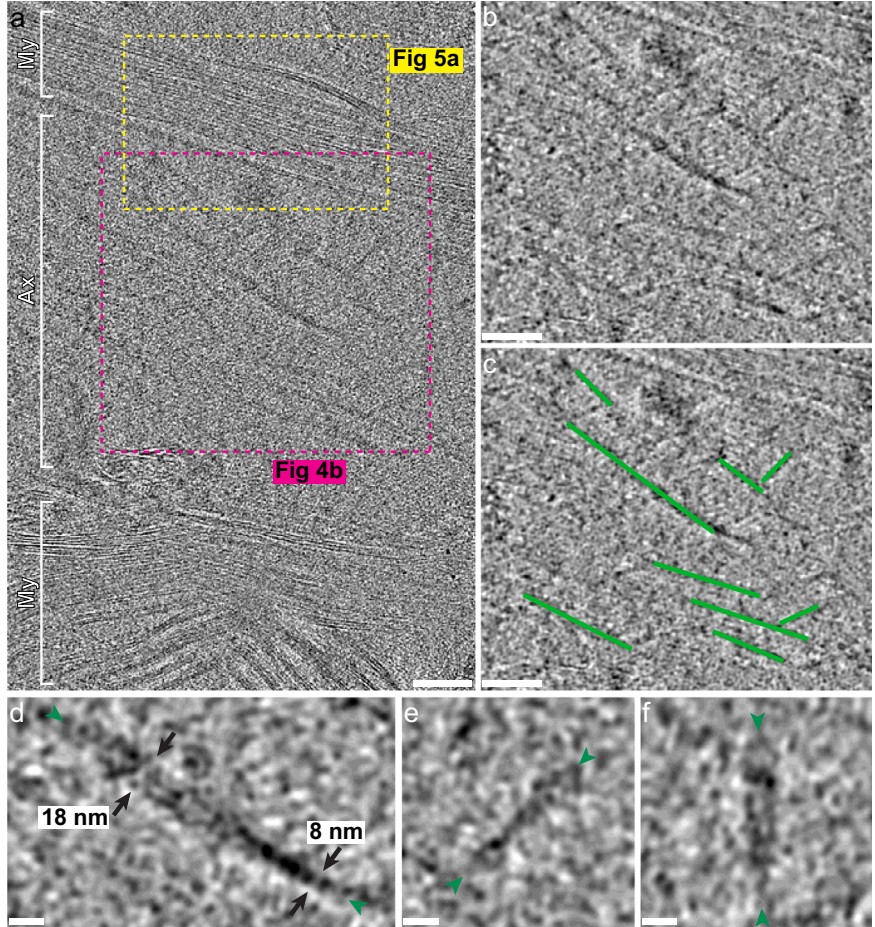

**Fig. 4 | Potential Alzheimer's disease tau is visualized within tomograms.**
**a** Cross-section of a tomogram without denoising showing a myelinated (My) axon (Ax) containing twisting filaments consistent with tau fibrils. Scale bar, 100 nm. **b** The pink box from **a** shown with denoising to highlight the potential twisting tau fibrils traced in (**c**). Scale bars, 100 nm. **d–f** Potential tau fibrils (ends shown with green arrowheads) with approximate maximum and minimum width shown (**d**, black arrows). Scale bars, 20 nm.

imaging and describing neurodegenerative disease aggregates with more native context than has previously been possible.

## Preservation of ultrafine myelin structures

One benefit of our approach is the ability to image difficult-to-recapitulate phenomena such as myelin sheaths in primary human tissue. Central nervous system myelin is an oligodendrocyte wrapping its processes around a neuronal axon[34,35]. Myelination is a complex event[35] that cannot be reliably reproduced in cell culture and requires tissue imaging. Even in 2D cryo-EM projection images, we visualized interesting myelin structures, including the inner tongue of oligodendrocytes (Supplementary Fig. 4c), which is thought to be important for continued myelin synthesis[35].

Tomograms revealed the 3D architecture of myelin, including the curvature, inter-weaving, and repetitive packing of myelin around axons (Fig. 3a, Fig. 4a, Fig. 5a, and Supplementary Movies 2, 3, and 5). Consistent with traditional EM imaging, in regions of compact myelin, we saw a repeated pattern of three parallel lines: two dark outside lines representing the extracellular facing phospholipid heads (dark blue or dark red in Fig. 5b) on either side of a dark and more dense internal line (light blue or light red in Fig. 5b), the major dense line, representing the two sets of intracellular facing phospholipid head groups and myelin basic protein (MBP) that holds myelin compact (Fig. 4a and Fig. 5).

In addition to the relatively consistent compact myelin, we saw areas of small cytoplasmic expansions corresponding to structures known as cytoplasmic channels (Fig. 5c, d). Cytoplasmic channels have

previously only been visualized by TEM using relatively low-resolution techniques (HPF then freeze substitution) in part because non-cryogenic methods of fixation and staining destroy these ultrafine structures[36]. Cytoplasmic channels are areas of structured oligodendrocyte cytoplasmic expansion where 2′,3′-cyclic nucleotide 3′-phosphodiesterase (CNP) associates with actin to counteract MBP allowing for intracellular transport from oligodendrocyte cell body to periaxonal space[36] since intracellular transport cannot efficiently cross orthogonally through compact myelin. In our images, we observed the junction of compact myelin and cytoplasmic channels as the major dense line splitting into two distinct lines resulting in a four-layered myelin structure (Fig. 5c, d).

We observed that the major dense line appeared to have a similar thickness to the lines of extracellular facing polar head groups (Fig. 5c), and at split points for cytoplasmic channels, the lines did not appear thinner once the intracellular facing polar head groups were separated (Fig. 5c, d). We performed subtomogram averaging of both compact myelin (Fig. 5e) and a cytoplasmic channel (Fig. 5g). We found that in regions of compact myelin, the extracellular facing polar head groups measured $1.76 \pm 0.02$ nm wide. Interestingly, the major dense line, representing two sets of intracellular facing polar head groups and MBP measured $1.47 \pm 0.03$ nm−less than the single sets of extracellular facing polar head groups ($p = 4 \times 10^{-7}$; Supplementary Fig. 8a). Furthermore, the subtomogram average of the region of a cytoplasmic channel showed that once the intracellular facing polar head groups split, they measured similar in width to the extracellular facing polar

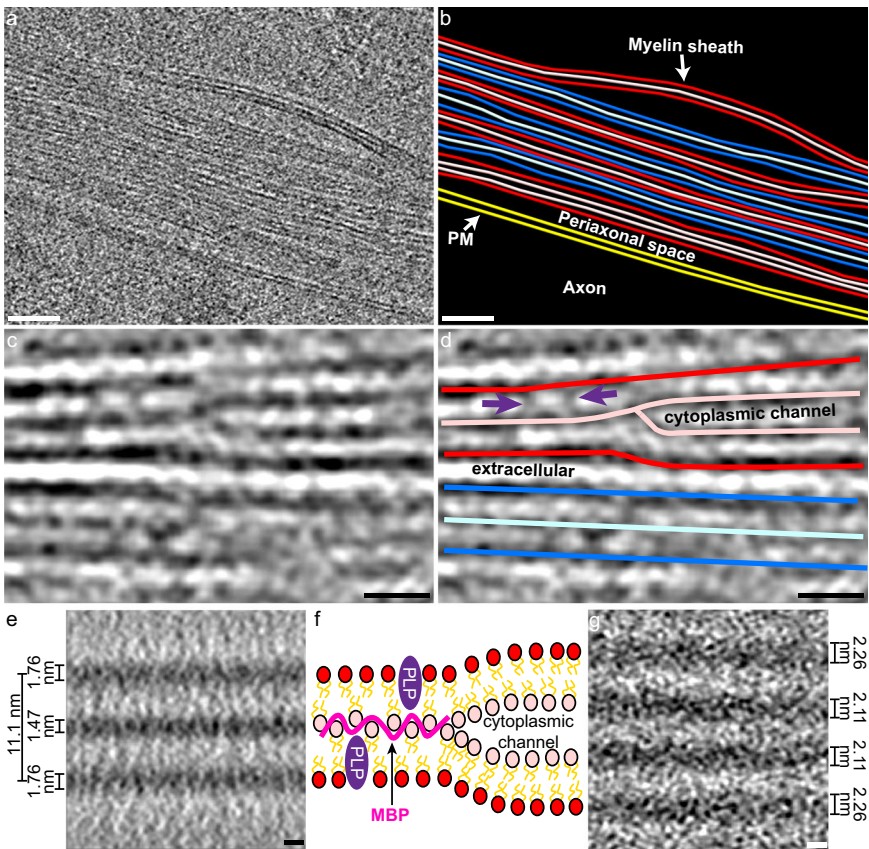

**Fig. 5 | Preservation of and insight into myelin architecture. a** Yellow box from Fig. 4a shows repeating myelination visualized by cryo-ET. **b** Segmentation of myelin shown in (**a**). Red and blue alternate layers of myelin, with darker shades indicating the extracellular surface and the lighter shades representing the intra-cellular surface of an oligodendrocyte. The layer of myelin closest to the axonal plasma membrane (PM; yellow) has a cytoplasmic channel—a gap between the intracellular surfaces of the oligodendrocyte. Scale bars, 50 nm for **a** and (**b**). (**c**) An example cytoplasmic channel where the major dense line splits into two separate electron-dense lines surrounding the expanded intracellular space. **d** A tracing of **c** (the blue line is extended as a model to where density can be seen again since the myelin is changing planes) with potential membrane protein densities (purple arrows). Scale bars, 10 nm for (**c**) and (**d**). **e** Subtomogram average of 4043 areas of compact myelin (average of 5 central Z slices with 2.675 Å voxel size). The total width of compact myelin is about 11.1 nm. The major dense line measures 1.47 ± 0.03 nm thick, which is less than the extracellular-facing polar head group densities that measure 1.76 ± 0.02 nm thick when averaged together. Scale bar, 2 nm. **f** Schematic of myelin architecture showing two sets of lipid bilayers with transmembrane myelin proteolipid protein (PLP) (purple) and myelin basic protein (MBP) (pink). MBP is shown to tightly bind to the intracellular facing polar head groups to rigidify and compact the two sets of intracellular polar head groups in compact myelin. Cytoplasmic channels form in the absence of MBP. **g** Subtomogram average of 127 areas of a cytoplasmic channel (average of 30 central Z slices with 2.675 Å voxel size). Scale bar, 2 nm.

head groups in the same region ($p = 0.33$; Supplementary Fig. 8b). These measurements from subtomogram averaging indicated that the presence of MBP pulls the intracellular polar head groups tightly together. It has been suggested that as a basic protein, MBP acts by tightly binding to the intracellular facing polar head groups of the oligodendrocyte plasma membrane and also that the polar head groups may serve to structure the intrinsically disordered MBP[37]. Our data suggest that MBP's tight association with lipid polar head groups holds the oligodendrocyte plasma membranes together, reducing the vertical motion of lipids within the membrane, thereby reducing the apparent thickness of the major dense line (Fig. 5f). Intracellular phospholipids found in cytoplasmic channels and all extracellular facing phospholipids are not constrained by MBP so will have more vertical motion, thereby increasing the apparent thickness of these layers compared to MBP-bound polar head groups (Fig. 5e and Supplementary Fig. 8). Based on the major dense line's thickness of 1.47 ± 0.03 nm, MBP likely poses as an elongated or flattened structure to make a tight mortar between the two sets of intracellular polar head groups (Fig. 5f).

Beyond our indirect observations of MBP, we observed densities that appear to span the myelin membrane layer (purple arrows Fig. 5d). Myelin has several transmembrane proteins, including myelin proteolipid protein (PLP) (Fig. 5f). These densities that span the lipid membrane bilayer may represent PLP and other transmembrane structural proteins that have been difficult to resolve via in vitro cryo-EM. Higher resolution and increased sampling may allow for sub-tomogram averaging to more definitively identify and structurally characterize these proteins.

## Discussion

Here we present a method for generating on-grid cryo-lamellae at variable heights in large tissue samples allowing for cryo-EM and cryo-ET visualization of human brain tissue from unfixed, never previously frozen samples acquired at the time of autopsy. We also present a method for vitrifying human brain tissue via plunge-freezing as opposed to HPF.

Previously, EM of human tissue has required fixation followed by sectioning and staining for visualization of features, a process that introduces artifacts and limits imaging resolution. Many human dis-eases and processes, particularly those of the brain, such as Alzhei-mer's disease and myelination, lack high-fidelity in vitro or animal models. Cryo-FIB milling followed by cryo-ET, while higher fidelity than traditional EM techniques, has mostly been performed on isolated cells in culture and has only recently moved toward imaging of tissue

samples[14,24]. Most tissue samples have been vitrified via HPF, which has lower throughput than plunge-freezing (~4 grids/hour with HPF[24] vs. ~20 grids/hour with plunge-freezing), frequently uses organic solvents that need to be sublimed, can embed samples in a block of ice, and requires technical expertise and equipment modification to freeze samples directly onto a cryo-EM grid. While plunge-freezing thick samples has been shown previously[14,27], our results indicate that different tissue samples require different protocols for vitrification, likely due to differences in lipid and water content. We anticipate our protocol may need adjustments for other tissue types.

Cryoprotectants from plunge-freezing do have potential artifacts as well. While glycerol has been co-opted as a frequent additive for vitrifying cells and, recently, tissues[14,27] and has shown minimal structural alterations for single-particle cryo-EM[38], it has the potential to cause membrane damage through osmotic effects. We do not see evidence of osmotic effects such as clear dehydration or membrane discontinuity. We do see reduced contrast in imaging, as has previously been described when using glycerol[38]. Trehalose has been less well-described for cryo-EM but is commonly used in nature to allow organisms to survive long periods of desiccation potentially through water replacement and maintenance of phospholipid bilayer integrity[21]. At near-atomic resolution, the replacement of water hydrogen bonding with trehalose may be consequential. However, inherent resolution limitations with tissue cryo-ET make these resolutions unlikely in the short term. Through its properties of hydrogen bond replacement, trehalose may indeed serve to counter the potential osmotic effects of glycerol.

Large samples also have been challenging for cryo-FIB milling due to milling rates. Gallium as an ion source is less precise at high beam currents and has been shown to interact with samples to alter their original structure[16], likely due to gallium's depth of sample interaction. Xenon plasma, however, has comparable beam characteristics to gallium at low currents, such as beam width at a focal point, but is significantly more precise at high currents and has a reduced interaction depth compared to gallium, making it suitable for large-volume samples that would take several days to thin by gallium-based FIB milling. Our protocol highlights how currents previously only used for material science applications can make large-volume FIB milling possible for biological specimens.

With this method, we generated thin lamellae in which we visualized the heterogeneous multicellular landscape of human brain tissue and subcellular structures within this milieu. We identified a diverse array of subcellular compartments, including putative endosomes, lysosomes, autophagic vesicles, granular vesicles, and phagophore-like structures. We also saw large proteinaceous structures in situ, including potential tau aggregates, that have, until recently, relied on sequential fractionation to extract and image ex vivo. Imaging of unfixed, never previously frozen tissue may allow for high-resolution confirmation of native tau aggregate structures as well as visualization and identification of the many different proteins that interact with tau aggregates in diseased brains. Our images also show myelin layers around an axon, a phenomenon that is not easily recapitulated in vitro, and we visualize areas key to normal myelin function. Future imaging of human tissue can describe in situ complex junctions such as the inner or outer tongue, nodes of Ranvier, or even areas of demyelination. We also visualized the major dense line of myelin and cytoplasmic channels at high enough resolution to determine that the two sets of intracellular polar head groups and MBP actually pack tighter than a single set of polar head groups without MBP, thereby further elucidating how MBP may work to keep myelin compact in vivo and how it may constrain phospholipid dynamics. Though myelin damage is a known process in Alzheimer's disease, the reported observations are likely physiologic as we do not see clear EM hallmarks of myelin degeneration[39] in the analyzed regions or in adjacent regions within the same lamellae. The observations highlighted also generally agree with previous observations of myelin but provide increased resolution by using cryo-EM.

Development of approaches, such as ours, that can attain high resolution with minimal artifacts can be of great value for making discoveries in the context of primary tissue. Benefits of our workflow include the use of never previously frozen tissue that should have better preserved cellular architecture, protein localization, and protein structure. Additionally, our method avoids the use of cryo-ultramicrotome sectioning to prevent cut artifacts and distortions. However, tissue acquired and vitrified at the time of autopsy also has drawbacks, including variable postmortem interval and perimortem events, unknown pathology type and localization at the time of freezing, and low availability of uncommon disease variants. Therefore, the development of several complementary approaches has value for studying human tissue as there are unique ethical and logistical challenges not present with cellular and animal models.

An inherent challenge of working with primary human brain tissue is postmortem interval. For the presented method, the limiting factor of this time is obtaining consent from the next of kin postmortem (this process is dependent on local regulations), transporting the individual to the morgue, and removing the individual's brain. Postmortem cells undergo a general stress response, including an increase in protein degradation and autophagy pathways, and have cell-type specific RNA transcript changes[40,41]. However, large mammalian brains exhibit an ability to retain metabolic and synaptic capabilities at least four hours postmortem[42]. Postmortem interval, though, is just one of many factors that influence tissue preservation. Perimortem events and disease processes also affect the condition of brain tissue at the time of death and soon thereafter. In the case of pre-frozen tissue banked in a tissue repository, these factors can be considered in choosing cases. Tissue vitrified at the time of autopsy, however, is currently not routine and thus is limited by donor availability and timing. In many cases, a full view of the donor's underlying pathophysiology is often not available until several weeks after tissue acquisition. These disadvantages are balanced by the significant advantage that tissue collected at the time of autopsy has never been frozen at −80 °C. In addition, different aspects of different cell types decompose at different rates postmortem under different conditions[4,43]. Therefore, the choice of approach depends on the biological question. Continued vitrification of human brain tissue at the time of autopsy over time may allow in the future for tissue selection from favorable cases for cryo-EM, like what can be done with brain tissue banked at −80 °C.

Our method provides a framework for imaging human phenomena in near-native environments that are not easily recapitulated in cell culture or animal models. This approach should be applicable to tissue samples thicker than what is reported here, but currently, it is limited by the height of the cartridge ring (~300 μm) used for automatic loading onto a Titan Krios. Our method could also be combined with existing techniques for correlative light and electron microscopy to allow the targeting of rare structures within tissue samples. While we present work in human brain tissue, this framework should be applicable to other human tissue, bulk tissue, or organoid samples. This method is an important step towards high-resolution cryo-EM imaging of both normal and diseased human tissue in the most native context possible.

## Methods

### Ethical statement

All autopsies were performed with informed consent from next of kin in accordance with local regulatory authorities. In the state of Pennsylvania, post-mortem studies are legally not considered to be human subjects research, and therefore, the University of Pennsylvania institutional review board does not provide formal oversight of these studies.

## Human tissue

Post-mortem brain tissue was obtained at the time of autopsy from brains donated to the Center for Neurodegenerative Research (CNDR) Brain Bank at the University of Pennsylvania. Informed consent was obtained from next of kin for all autopsies in accordance with local regulatory authorities. The illustrative cryo-ET sample highlighted in this manuscript was obtained from a man in his 80s with a 13-year history of behavioral variant frontotemporal dementia and a post-mortem interval of 28 hours. Neuropathologic examination revealed a high level of Alzheimer's disease neuropathologic change (A3, B3, C3), Braak VI, moderate cerebral amyloid angiopathy, and severe arteriosclerosis. Severe tau and amyloid burden were seen in the middle frontal cortex. No TDP-43 or α-synuclein aggregates were seen.

## Sample preparation

Human brain tissue was acquired at the time of autopsy from the middle frontal cortex. Tissue was acquired from autopsies performed between 7 and 28 h postmortem. After the acquisition, the tissue was kept on ice. Sections of cortex less than 3 mm wide were isolated using pre-cooled razor blades on ice. Tissue sections were embedded in cool 6% low-melting point agarose (ThermoFisher, Waltham, MA) in DPBS. Tissue embedded in agarose was cooled on ice for 15 min. The agarose-embedded tissue was then superglued to the vibratome stage and allowed to dry on ice for at least 15 min. Next, the vibratome stage with attached tissue was inserted into the vibratome (Model G Oxford Vibratome or Leica VT1200S, Leica Biosystems, Deerfield, IL, USA), surrounded by fresh ice, and then filled with DPBS. Tissue was cut with the vibratome set to 100 μm sections using a sapphire blade (Ted Pella, Redding, CA, USA), maximum amplitude, speed 2 (Oxford) or 0.5 mm/ s (Leica), and blade angle slightly below 0° (Oxford) or -15° as measured by vibratome guide markings (Leica). As tissue sections were cut, they were moved to cold DPBS, then cold 20% glycerol with 1 M trehalose in DPBS solution for 70 min. Tissue sections were then placed onto 200 mesh or 100 mesh bare copper grids (Electron Microscopy Sciences, Hatfield, PA, USA), manually blotted from behind the grid, and plunge-frozen into a liquid ethane/propane mixture using a Leica GP2 (Leica Microsystems, Deerfield, IL, USA). Frozen grids were then clipped into autogrids (ThermoFisher).

## FIB milling

Xenon plasma FIB milling was performed on a Tescan S8000X FIB/SEM with Leica VCT and VCM operating at 30 keV beam energy. The stage was cooled to cryogenic conditions. Prior to loading grids onto the FIB/SEM, the beams were manually checked and tuned on silicon. The gas injection system organometallic platinum was warmed to 70 °C. Gas injection system platinum was expelled for two min prior to loading sample grids. Grids were transferred to the FIB/SEM using a Leica VCT and VCM under cryogenic conditions. The SEM beam was used to assess grid quality and orientation prior to platinum coating. Grids were oriented so the FIB beam would be close to orthogonal to horizontal grid bars. After an initial assessment, grids were coated with a gas injection system platinum for 1 min total in 15 s increments with 15 s rest intervals between injections. This initial gas injection system platinum deposition was carried out at −30° rotation, 0° tilt. The organometallic platinum layer was then cured by rastering the FIB beam over the full tissue sample three times using 10 nA, 30 keV at the field of view (FoV) 1200.7 μm with raster speed 4 at −30° rotation, 10° tilt.

Trench milling was performed to reveal an internal tissue face. Trench milling was performed at −30° rotation, 5° tilt, with the FIB nearly orthogonal to the plane of the grid (15° being orthogonal). Initial milling steps at 1 μA and 100 nA used box milling patterns, and all milling steps below 100 nA used directional polishing patterns. Milling was initiated at 1 μA current until about 80–100 μm from the targeted finishing location of trench milling. Targeted trench finishing was located just behind a grid bar to allow for maximum imaging area and prevent grid bars from occluding transmission EM imaging. 100 nA polishing pattern was used to mill up to the grid bar directly in front of future lamellae positions. 100 nA square pattern was then used to mill away grid bars in front of future lamellae positions. 10 nA current with a polishing pattern was used to smooth curtaining at all spots of potential lamellae.

After trench milling, the gas injection system organometallic platinum was deposited onto the exposed tissue face with a retracted gas injection system needle at −165° rotation, 40° tilt (gas injection system needle orthogonal to the tissue face). Curing of this platinum layer using the FIB beam was performed at −150° rotation, 20–27° tilt depending on the field of view so that all areas of platinum covering tissue to be milled were cured. Curing was performed as described above. Gas injection system platinum was deposited for 5 min in total, with iterative depositions alternating with curing by the following pattern: 15 s deposition, cure, 15 s deposition, cure, 30 s deposition, cure, 30 s deposition, cure, 30 s deposition, cure, 1 min deposition, cure, 1 min deposition, cure, 1 min deposition, cure.

Lamellae were milled at −150° rotation with a final tilt angle of between 19° and 23°. Expansion joints were made on either side of the lamellae using 10 nA beam to roughly 8 μm wide. The 10 nA beam was also used to thin the tissue to a thickness of 10–20 μm. Material was first removed from below the intended lamella position from an initial tilt angle of 21–27°. The tilt angle was then reduced incrementally when possible as material was removed from the underside of the tissue. After excess material from beneath the final lamella was removed, the milling tilt angle was adjusted to −1° relative to the final intended lamella angle (i.e., if the final lamella was intended to be 19°, then milling from below was performed at 18°). The material was removed above the intended lamella position at +1° from the final intended lamella angle, also using the 10 nA beam. Starting at 10–20 μm thickness, over and under tilting was reduced to ±0.5°, and the beam current was reduced to 1 nA. Around 1–3 μm thickness, over and under tilting was reduced to ±0.3°, and the beam current was reduced to 100 pA. Final polishing steps were carried out at 100 pA and then 50 pA. This final polishing step was carried out on all lamellae on a given grid just prior to the removal of the grid from the FIB/SEM to ensure minimal surface ice deposition. The lamellae were polished to a final thickness of <400 nm, as measured by FIB images. Lamella grids were unloaded under cryogenic conditions using Leica VCT and VCM and stored in liquid nitrogen until they were imaged via transmission EM.

## Cryo-EM

Cryo-EM was performed on a Thermo Fisher Krios G3i 300 keV field emission cryo-transmission electron microscope. Imaging was performed using the SerialEM 4.1 software[44] on a K3 direct electron detector (Gatan, Pleasanton, CA, USA) operated in electron-counted mode. Imaging was performed with a slit width of 50 eV between magnifications of ×4800 and ×19,500. Imaging at magnification of ×33,000 was performed with a slit width of 20 eV. Imaging at magnification of ×82 and ×470 was used to assess grid orientation to ensure lamellae were loaded orthogonal to the tilt axis and assess suitability for imaging after transfer. Further montages were taken at magnifications ×4800 and ×19,500 to identify features in lamellae that would be suitable for high-magnification imaging. Tilt series were collected with a span of 72° (±36° from the tilt angle at which lamellae were orthogonal to the electron beam; dose-symmetric scheme) with 3° increments at a magnification of ×33,000 (with a corresponding pixel size of 2.675 Å) and a defocus target of −6 μm. Tilt series were collected with a cumulative dose of between 86 and 97 e⁻ Å⁻². 

with a cumulative dose of between 86 and 97 $e^- \text{Å}^{-2}$.

## Image processing

Montages at a magnification of ×19,500 were rotated to align curtaining with the vertical axis of the image. Similar to a recently

published procedure[14], a vertical filter was applied with 95% tolerance of direction in Fiji[45] to remove curtaining artifacts.

Data processing of tilt series is summarized in Supplementary Fig. 9. Similar to montage processing, tilt series were first filtered in Fiji using a high-pass filter of 1000 pixels and a vertical filter with 95% tolerance of direction. Tilt series were aligned using platinum deposition on the surface of the lamellae as fiducials and reconstructed into tomograms via weighted back projection in IMOD 4.11.24[46]. Tomograms were reconstructed with an exact filter of 90 unbinned pixels or a SIRT-like filter of 10 iterations. For visualization, some tomograms were denoised using 3D denoising with or without a Gaussian filter in Topaz 0.2.5[47].

For subtomogram averaging of compact myelin, model points were placed along straight regions of the major dense line using IMOD in 4× binned (10.70 Å voxel size) tomograms. Additional model points were added every 2 voxels using 'addModPts' command in PEET 1.16.0a[48] for a total of 4043 model points. PEET was used for subtomogram averaging in unbinned tomograms (2.675 Å voxel size) with iterations of rotational and translational searching to align the different regions of myelin with a box size of $96 \times 96 \times 96$. Fiji was used to define values of the average intensity for five adjacent central sets of 10 sequential voxels in $Z$ of the subtomogram average ranging from $Z = 20$ to $Z = 70$. From these values, the full-width half maximum was used to measure the width of the major dense line and extracellular facing polar head groups by fitting the voxel intensity over each line to a Gaussian curve. In the average of compact myelin, the two outer lines were grouped together. A Student's $t$-test, with a two-tailed distribution, was used to determine the significance of the full width half maximum measurement for inside dark lines compared to outside dark lines. A similar method was employed for subtomogram averaging of cytoplasmic channels and subsequent measurement of polar head group density, with model points being placed at the center of a cytoplasmic channel for a total of 127 model points. For the cytoplasmic channel average, the two outer lines were grouped together, and the two inner lines were grouped together. The width of the total myelin repeating unit was measured from maximum to maximum of the extracellular facing lipid head densities.

Manual segmentation/tracing of myelin components was produced in IMOD and imported into ChimeraX 1.6.1[49] for visualization and figure generation.

## Immunohistochemistry

During vibratome tissue sectioning, pieces of tissue immediately adjacent to those saved for cryo-EM were fixed in 10% neutral buffered formalin. Tissue was embedded in paraffin blocks and cut into 6-μm thick sections. Well-characterized primary antibodies to detect tau (PHF1 provided by Peter Davies)[50], amyloid-β (NAB228 provided by Center for Neurodegenerative Disease Research at the University of Pennsylvania)[51], TDP-43 (1D3 provided by Elisabeth Kremmer and Manuela Neumann)[52], and α-synuclein (SYN303 provided by Center for Neurodegenerative Disease Research at the University of Pennsylvania)[53] were used as previously described[54]. Slides were deparaffinized in xylene and then descending ethanol concentrations (100%, 90%, 80%, 70%). For NAB228, 1D3, and SYN303, slides were placed in water, then 88% formic acid epitope retrieval buffer for 5 min, followed by a 10 min wash in water. All slides were then immersed in 5% $H_2O_2$ for 30 min for peroxidase quenching and then washed in water for 10 min. Slides were washed in 0.1 M Tris pH 7.6 for 5 min, then blocked in 0.1 M Tris pH 7.6 with 2% fetal bovine serum. Slides were then incubated for 16 hours with primary antibodies (1:2000 for PHF1, 1:300 for 1D3, 1:10,000 for SYN303, and 1:30,000 for NAD228). After incubation, slides were washed with 0.1 M Tris pH 7.6 and then blocked in 0.1 M Tris pH 7.6 with 2% fetal bovine serum. Binding was then detected using species-specific biotinylated secondary antibodies and developed with Vectastain Avidin-Biotin Complex kit (Vector Laboratories) and ImmPACT 3'-Diaminobenzidine (Vector Laboratories). Slides were counterstained with Harris's hematoxylin. TMEM239 primary antibody (gift from Michel Goedert) was used to detect TMEM106B[55]. A similar approach was used for the TMEM239 antibody, except with distinct buffers. The buffers used were 10 mM Tris, 1 mM EDTA, 0.05% Tween-20, pH 9.0 for epitope retrieval, 3% hydrogen peroxide, and 20% methanol in PBS for peroxidase quenching, 0.3% Triton X-100 in PBS for wash steps, and 2.5% bovine serum albumin, 5% fetal bovine serum in 0.3% Triton X-100 in PBS for blocking. TMEM239 was used with 1:500 dilution.

## Reporting summary

Further information on research design is available in the Nature Portfolio Reporting Summary linked to this article.

## Data availability

The tomograms shown in this study have been deposited in the Electron Microscopy Data Bank (EMDB) under accession codes: EMD-43334 (Fig. 2d), EMD-43330 (Fig. 3), and EMD-43331 (Fig. 4). The raw tilt series for the tomograms shown in this study have been deposited in the Electron Microscopy Public Image Archive (EMPIAR) under accession code EMPIAR-11920. Source data are provided with this paper.

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

## Acknowledgements

We are grateful to the patients, families and caregivers that make this research possible. We thank Jamie Ford at the Singh Center for Nanotechnology and Stefan Steimle at the Beckman Center for Cryo-Electron Microscopy at the University of Pennsylvania for their assistance. This research was partially supported by the National Science Foundation through the University of Pennsylvania Materials Research Science and Engineering Center (DMR-2309043). We thank Michel Goedert for graciously providing the TMEM239 antibody. We also thank Elisabeth Kremmer and Manuela Neumann for providing the 1D3 (anti-TDP-43) antibody. This study was supported by the DeCrane Family Fund for PPA Research, a gift from the Shanahan Family Foundation, and grants RF1AG065341 to E.B.L. and Y.-W.C., P30AG072979, P01AG066597, and U19AG062418 from the National Institute of Health to E.B.L.; a David and Lucile Packard Fellowship for Science and Engineering (2019-69645),

Burroughs Wellcome Fund Investigators in the Pathogenesis of Infectious Disease Program (1022785), a Pennsylvania Department of Health FY19 Health Research Formula Fund, and RM1GM136511 from the National Institute of Health to Y.-W.C.; and R35GM130302 from the National Institute of Health to B.E.B. Training support was provided by the National Institute of Health T32GM132039 and F30AG077756 to B.C.C., and F30CA261198 to K.K.

## Author contributions

B.C.C., K.K., E.B.L., and Y.-W.C. conceived the study and designed the experiments. B.C.C. and K.K. performed the experiments. B.C.C., K.K., E.B.L., and Y.-W.C. analyzed the data. B.E.B. provided critical input on the study. The manuscript was written by B.C.C., K.K., E.B.L., and Y.-W.C. with input from all the authors.

## Competing interests

The authors declare no competing interests.
