## [Peer Review File · Nature Communications]

Reviewers' Comments:

Reviewer #1:

Remarks to the Author:

The manuscript from Creekmore et al, describe both a method to prepare and image large tissue using cryo electron tomography and describes the cellular ultrastructure changes visible in Alzheimer patients.

I have a fundamental problem with the article, the way it is presented, which is in principle presenting a method but then spending a significant portion of the content on the biological results. The work is fantastic, but I believe a choice should be made with either a much stronger methodological focus, and leave the biological results aside or the biology should be addressed with a better comparative analysis with non-diseased tissue. The direction to go is an editorial choice, mine is a suggestion.

I am very surprised that heat damage is not visible after milling at 1uA.

Could the authors perhaps provide evidence of how close the various currents can be used to produce good and undamaged samples? The approach is very sensible and the numbers presented are likely conservative to ensure a result, but I would ask for a "titration curve" for the distances. From direct experience, We had attempted to mill at currents higher than 200nA and have always faced the charging issues, can the authors better describe how they get around those, and what is the minimum metal coat compatible with the high current? the problem we saw occurring most often is a significant distortion of the milling pattern due to the deposited charge, it is extremely interesting to see this issue disappears in this work.

one minor point, I find the introduction a little too long and dispersive.

other than that this work should be published when the focus has been improved and the above description of the current use has been addressed.

regards

Alex de Marco

Reviewer #2:

Remarks to the Author:

This study pioneers the use of cryo-electron tomography to image fresh human autopsy brain tissue. This is a substantial technical achievement that has huge potential to reveal novel biological insights, especially into the poorly-understood molecular pathology of neurodegenerative diseases such as Alzheimer's. Accordingly, I enthusiastically support the publication of this work. That being said, there are a number of points, outlined below, that I believe should be addressed prior to publication in order to strengthen the manuscript.

1. The samples have long post-mortem intervals of 7 to 28 h and high concentrations of cryo-protectants, such as 1 M trehalose. It would be helpful if the authors could add a short discussion about possible artefacts arising from these two details, similar to their discussion about the use of tissues that were previously frozen in other studies. In addition, the post-mortem interval of the specific case examined should be clearly stated.

2. I would not have expected that 100 micron-thick sections could be uniformly vitrified by plunge-freezing. It would be helpful to report the number of lamellae examined, what the success rate of vitrification was, and how uniform this was. In addition, Extended Data Figure 1 shows a projection image of a large field of view for the lamella containing crystalline ice, but only single tilt images of smaller fields of view for the lamella that does not contain vitreous ice, which is not a fair comparison. It would also be helpful to include additional analysis to test for the presence of crystalline ice, such as fast Fourier transforms of the images.

3. The analysis used to support the identification of tau fibrils is not convincing. The authors should

measure and report the helical cross-over distances of the fibrils where possible. Are these consistent with tau fibrils? Intracellular TMEM106B fibrils are also abundant in aged human brains (Schweighauser et al. 2022. Nature 605 310-314). Could the observed fibrils be composed of TMEM106B? This should be discussed. In addition, it would be useful to show the data that supports the claim that the sample contained tau and amyloid-beta deposits, but not those of TDP-43 and alpha-synuclein.

4. It would be helpful if the authors could report if they observe any interactions between the putative tau filaments and the cellular milieu. Such interactions may contribute to filament formation and/or cytotoxicity, and so would be of great interest to the field.

5. It would be useful to include a discussion on whether the observations relating to myelin ultrastructure reflect physiology or pathology, considering that the tissue studied exhibited Alzheimer's disease neuropathologic change and damage to myelin is a pathological hallmark of Alzheimer's disease.

6. The Results text jumps between the use of past and present text. The manuscript would be easier to follow if the results were all written in the past tense, as is standard.

7. The sentence referring to the study by Gilbert et al. in BioRxiv beginning line 345, 'That study obtained novel subtomogram averages of in situ tau fibrils, which raise intriguing questions about differences between in situ and ex vivo tau fibril averages,' requires clarification. In that study, the authors were able to fit ex vivo tau fibril structures into all of their low-resolution sub-tomogram averages, without any noticeable differences.

Reviewer #3:

Remarks to the Author:

The manuscript describes a novel approach to plunge-freeze human tissue samples, and to perform subsequent plasma-FIB milling and cryo-ET. This is one of the first examples where it is shown to be possible to obtain fresh human tissue samples, and vitrify them in an accessible way by plunge-freezing. The authors describe a protocol for how to achieve the vitrification of thicker tissue samples (~100-200 um), and also describe in detail how to create lamellae with the help of plasma FIB milling. This manuscript is definitely interesting for the general audience. The manuscript does show the applicability of this new sample preparation method. It is impressive to see that a biopsy of a human brain can be vitrified by just plunge freezing and turned into lamellae, and then imaged by cryo-ET.

A few minor changes could further improve the manuscript:

- Ext. data figure 3A and Figure 2A do look like there is some crystalline ice (dark shadows). Could you please comment whether those come just from ice contamination on top of the lamellae, or if they are areas of local non-vitreous ice (and how often this was encountered)? To show if this is contamination or not, could you please show those (or similar) example slices alongside with the xz and/or yz slices through the tomogram?

- The introduction (Lines 39-80) should be more concise. The concepts important to the paper should be introduced in a more succinct way.

- Please use 3D for Three-dimensional and 2D for Two-dimensional both in the text in figure legends for the sake of space and clarity.

- The methods section says "Tilt series were aligned using platinum deposition on the surface of the lamellae as fiducials and reconstructed into tomograms via weighted back projection in IMOD". Some of the tomographic reconstructions look like the tilt series alignment was a bit suboptimal (videos 1-2). Could you please use patch tracking (with minimal number of patches 2-4, or even just coarse alignment!), and see if the reconstruction will look better? alignment based on platinum fiducials often skews the center of the tomogram where your features of interest are. If

it's better, then please show the new reconstructions.

- Could you please report on the "statistics": what was the number of brain samples that you obtained in total, how many lamellae were produced? How much FIB-SEM time did it take approximately? How many lamellae were in the end imaged? And how many successful tomograms were acquired? This is important to know for someone who might like to use your new approach.
- What was the average lamellae thickness? Please report this in the methods section or in the results section.
- Could you please show a segmentation of the myelin membranes in one of the figures/videos? This would really improve the visual read out of the paper. Since most observations are descriptions, the 3D segmentation could really aid the reader.

REVIEWER COMMENTS

Reviewer #1 (Remarks to the Author):

The manuscript from Creekmore et al, describe both a method to prepare and image large tissue using cryo electron tomography and describes the cellular ultrastructure changes visible in Alzheimer patients.

I have a fundamental problem with the article, the way it is presented, which is in principle presenting a method but then spending a significant portion of the content on the biological results. The work is fantastic, but I believe a choice should be made with either a much stronger methodological focus, and leave the biological results aside or the biology should be addressed with a better comparative analysis with non-diseased tissue. The direction to go is an editorial choice, mine is a suggestion.

We sincerely appreciate Dr. De Marco's thoughtful and constructive suggestion regarding our manuscript style. Our aim is to showcase the methodological innovation while also acknowledging the significance of the biological findings. We understand the concern raised about the balance between the two aspects. In response, we have dedicated considerable effort to strengthen the methodological focus in the revised manuscript. We have incorporated additional methodological details and statistics, as outlined in our responses to the reviewers' comments. This will enhance the clarity and depth of the methodological presentation. Furthermore, we have streamlined the presentation of biological results, using them judiciously as illustrative examples to underscore the capabilities of the method. This approach allows us to maintain a strong focus on the methodology while providing valuable insights into the biological implications of our study.

I am very surprised that heat damage is not visible after milling at 1uA.

Could the authors perhaps provide evidence of how close the various currents can be used to produce good and undamaged samples? The approach is very sensible and the numbers presented are likely conservative to ensure a result, but I would ask for a "titration curve" for the distances.

We appreciate Dr. De Marco's request for additional information on the "titration curve" experiment. We diligently attempted to conduct this experiment by employing different currents of the focused ion beam for trench milling orthogonal to the grid plane in preparation for generating lamellae suitable for cryo-TEM examination. However, we encountered significant challenges that hindered the reliable execution of the experiment.

During trench milling if we attempted to complete the full process with currents >10 nA, we observed a severe curtaining effect that created deep fissures in the milled front surface of the tissue. These deep curtains resulted in poor downstream coating of platinum on the trench front surface and hindered proper protection during lamellae generation through subsequent low-angle milling. As a consequence, this made a true "titration curve" impossible as we were unable to reliably obtain lamellae when the surface was not smoothed out by 10 nA beams. The depth of the curtaining, which can vary by sample-specifics, creates a limit on how close the high current (>10 nA) beams can be used.

In addition, the tissue pieces frozen on the grid exhibited irregular contours and varying thicknesses across. This irregularity posed a challenge during trench milling, particularly when comparing different currents. The amount of material needed to be removed to create each trench front varied due to the irregular shape and thickness of the frozen samples.

Consequently, rigorously comparing distances at which a current is used is difficult as there is variability from trench to trench on how long a beam is needed at each point.

As rightly pointed out by Dr. De Marco, we opted for a conservative approach in the milling procedure. This decision aimed at minimizing effects from sample irregularities and prioritizing the preservation of sample integrity to enable reliable and consistent lamellae generation for subsequent cryo-TEM examination.

From direct experience, We had attempted to mill at currents higher than 200nA and have always faced the charging issues, can the authors better describe how they get around those, and what is the minimum metal coat compatible with the high current? the problem we saw occurring most often is a significant distortion of the milling pattern due to the deposited charge, it is extremely interesting to see this issue disappears in this work. We thank Dr. De Marco for generously sharing their experience.

Interestingly, during trench milling orthogonal to the grid plane, we have observed minimal distortion caused by charging, even when utilizing a 1 μ A or 100 nA beam, coupled with a modest amount of Pt deposition. However, during low-angle lamella milling, we have also noticed clear distortion in the milling pattern, particularly when using currents higher than 10 nA. We overcame this issue by determining that the manner in which Pt is deposited, rather than the quantity deposited, plays a crucial role in mitigating distortion, and charging in general, in our setup.

Our observations indicate that when Pt is deposited to the trench front at once without intermittent curing using the ion beam, the resulting Pt layer tends to be porous. This porous nature likely does not effectively conduct electrons nor shield the lamellae against high-current beams, leading to significant charging issues, damage to the lamella, and unreliable lamella thinning. To overcome this challenge, we have implemented an iterative curing method in this study. This method facilitates the deposition of a more solid and compact Pt layer on the trench front, reducing charging and ensuring adequate protection of the lamella when milling.

However, despite these measures, we have found that the use of a 100 nA beam during the low-angle lamella milling step is not entirely predictable and can still result in pattern distortion. Additionally, even using the 100 nA beam >30 μ m from the location of the final lamella caused persistent charging of the tissue that was incompatible with reliable lamella thinning. Consequently, we have made the decision to exclude the use of the 100 nA beam entirely during the lamella milling step, opting instead for currents of 10 nA and lower in the finalized procedure.

To enhance the clarity of our methodology, we have incorporated additional text and figure in the manuscript (lines 208-212, 241-253, 278-288, Supplementary Fig. 3) to explicitly describe the rationale behind our approach, emphasizing the importance of Pt deposition methodology in mitigating charging issues during the milling process.

one minor point, I find the introduction a little too long and dispersive.

We appreciate this comment and have revised the introduction to be more streamlined and focused.

other than that this work should be published when the focus has been improved and the above description of the current use has been addressed.

regards
Alex de Marco

Reviewer #2 (Remarks to the Author):

This study pioneers the use of cryo-electron tomography to image fresh human autopsy brain tissue. This is a substantial technical achievement that has huge potential to reveal novel biological insights, especially into the poorly-understood molecular pathology of neurodegenerative diseases such as Alzheimer's. Accordingly, I enthusiastically support the publication of this work. That being said, there are a number of points, outlined below, that I believe should be addressed prior to publication in order to strengthen the manuscript.

1. The samples have long post-mortem intervals of 7 to 28 h and high concentrations of cryo-protectants, such as 1 M trehalose. It would be helpful if the authors could add a short discussion about possible artefacts arising from these two details, similar to their discussion about the use of tissues that were previously frozen in other studies. In addition, the post-mortem interval of the specific case examined should be clearly stated.

We thank the reviewer for these suggestions. We have now added paragraphs to the discussion about postmortem effects and considerations for tissue (lines 545-568) and potential effects of trehalose and glycerol (lines 488-499).

(lines 545-568) "An inherent challenge of working with primary human brain tissue is postmortem interval. For the presented method, the limiting factor of this time is obtaining consent from next of kin postmortem (this process is dependent on local regulations), transporting the individual to the morgue, and removing the individual's brain. Postmortem cells undergo a general stress response, including an increase in protein degradation and autophagy pathways, and have cell-type specific RNA transcript changes^{40,41}. However, large mammalian brains exhibit an ability to retain metabolic and synaptic capabilities at least four hours postmortem⁴². Postmortem interval is just one of many factors that influence tissue preservation though. Perimortem events and disease processes also affect condition of brain tissue at the time of death and soon thereafter. In the case of prefrozen tissue banked in a tissue repository, these factors can be considered in choosing cases. Tissue vitrified at the time of autopsy, however, is currently not routine, and thus is limited by donor availability and timing, and a full view of the donor's underlying pathophysiology is often not available until several weeks after tissue acquisition. These disadvantages are balanced by the significant advantage that tissue collected at time of autopsy has never been frozen at -80 °C. In addition, different aspects of different cell types decompose at different rates postmortem under different conditions^{4,43}. Therefore, choice of approach depends on the biological question. Continued vitrification of human brain tissue at time of autopsy over time may allow in the future for tissue selection for cryo-EM like what can be done with brain tissue banked at -80 °C."

(lines 488-499) "Cryoprotectants from plunge-freezing do have potential artifacts as well. While glycerol has been co-opted as a frequent additive for vitrifying cells and recently tissues^{14,27} and has shown minimal structural alterations for single-particle cryo-EM³⁸, it has the potential to cause membrane damage through osmotic effects similar to trehalose or other cryoprotectants. We do not see evidence of osmotic effects by clear dehydration or membrane discontinuity. We do see reduced contrast in imaging as has previously been described when using glycerol³⁸. Trehalose has been less well-described for cryo-EM, however is used naturally by many organisms to survive long periods of desiccation potentially through water replacement and maintenance of phospholipid bilayer integrity²¹. At near-atomic resolution, replacement of water hydrogen bonding with trehalose may be consequential, however, inherent resolution limitations with tissue cryo-ET make these resolutions unlikely in the short-term. Through its properties of

hydrogen bond replacement, trehalose may indeed serve to counter the potential osmotic effects of glycerol.”

We have now added the post-mortem interval of tissues used for illustrative tomograms to the methods section at line 591.

2. I would not have expected that 100 micron-thick sections could be uniformly vitrified by plunge-freezing. It would be helpful to report the number of lamellae examined, what the success rate of vitrification was, and how uniform this was. In addition, Extended Data Figure 1 shows a projection image of a large field of view for the lamella containing crystalline ice, but only single tilt images of smaller fields of view for the lamella that does not contain vitreous ice, which is not a fair comparison. It would also be helpful to include additional analysis to test for the presence of crystalline ice, such as fast Fourier transforms of the images.

We have generated and examined a total of 17 lamellae under the cryo-protectant conditions of 1M trehalose and 20% glycerol, with an average xy area of approximately 1900 μm^2 per lamella. All of these lamellae exhibited no evidence of crystalline ice. While we cannot entirely rule out the possibility that some areas within the tissue may be non-vitreous but were milled away during lamellae generation, our examination provides robust evidence of effective vitrification of these relatively thick tissue sections. These statistics are now reported in the revised manuscript (Lines 164-166).

In response to the reviewer's valuable recommendation, we have included an expanded field of view image of a vitreous lamella under the 20% glycerol + 1M trehalose condition (Supplementary Fig. 1b). This image is now displayed alongside the one previously presented for the non-vitreous 10% glycerol condition (Supplementary Fig. 1a), allowing for a direct and fair comparison.

Furthermore, to provide a comprehensive comparative analysis, we have introduced tilt-series images of both the vitrified lamella and a sample containing crystalline ice. In the tilt-series images of the sample with crystalline ice (Supplementary Fig. 2a,c,e,g), the presence of Bragg reflections caused the crystalline ice to exhibit changing contrast at different tilt angles, presenting a distinctive pattern from the vitreous background. Conversely, the tilt-series images of the lamella generated by our finalized method (Supplementary Fig. 2i,k,m,o) do not display such features, indicating an absence of crystalline ice and suggesting full vitrification.

In accordance with the reviewer's suggestion, we have also performed fast Fourier transforms (FFT) of these tilt-series images to inspect potential signals of crystalline ice (Supplementary Fig. 2b,d,f,h for the non-vitreous tilt series and Supplementary Fig. 2j,l,n,p for the tilt series generated with the presented method). The FFTs for the non-vitreous tilt series do show dots potentially indicative of crystalline ice diffraction. However, we acknowledge that none of these FFTs exhibit a very clear crystalline diffraction pattern, possibly due to low electron dosage or low magnification during the acquisition of the tilt-series data. While FFTs may not serve as a definitive metric of vitreous ice in this specific context, these analyses collectively support the conclusion that plunge-freezing with suitable cryo-protectants can effectively vitrify brain tissue sections of such thickness.

3. The analysis used to support the identification of tau fibrils is not convincing. The authors should measure and report the helical cross-over distances of the fibrils where possible. Are these consistent with tau fibrils? Intracellular TMEM106B fibrils are also abundant in aged human brains (Schweighauser et al. 2022. Nature 605 310-314). Could the observed fibrils be composed of TMEM106B? This should be discussed. In addition, it would be useful to show the

data that supports the claim that the sample contained tau and amyloid-beta deposits, but not those of TDP-43 and alpha-synuclein.

We thank the reviewer for this astute point. In response, we retrieved the remaining vibratome sections from the exact same piece of brain tissue used for the study (we always save the leftover vibratome sections via chemical fixation for long-term archiving). We performed immunostaining on these sections and confirmed that the regions adjacent to the cryo-ET-imaged area were positive for tau, amyloid- β , and TMEM106B, while being negative for α -synuclein and TDP-43. These new data are now reported in the new Supplementary Figure 5.

In response to the suggestion on image analysis, we have now measured the helical cross-over distances of fibrils where possible (Supplementary Fig. 6). The measured crossover distances and fibril dimensions appear to be more consistent in most instances with AD tau fibrils, although we cannot entirely rule out the presence some TMEM106B fibrils. We have updated our discussion of our findings with these new data (lines 361-369).

4. It would be helpful if the authors could report if they observe any interactions between the putative tau filaments and the cellular milieu. Such interactions may contribute to filament formation and/or cytotoxicity, and so would be of great interest to the field.

We acknowledge the importance of investigating interactions between putative tau filaments and the cellular milieu, as such interactions could play a crucial role in filament formation and cytotoxicity. In our current reported data, with the resolution limitations and small number of images of filaments, we did not observe clear interactions between the putative tau filaments and the cellular environment. However, we emphasize that the capability to observe such interactions is a critical aspect of in situ cryo-ET, and it has been a major motivation for establishing this method.

We anticipate that future extensive studies using this new method on a broader range of samples will provide opportunities to observe and analyze in situ interactions. We are committed to exploring these aspects in our future investigations, and we agree that understanding these interactions is a key direction for advancing our understanding of filament formation, cytotoxicity, and related phenomena in neurodegenerative diseases. We appreciate the reviewer's insightful suggestion and look forward to contributing to this important avenue of research in subsequent studies.

5. It would be useful to include a discussion on whether the observations relating to myelin ultrastructure reflect physiology or pathology, considering that the tissue studied exhibited Alzheimer's disease neuropathologic change and damage to myelin is a pathological hallmark of Alzheimer's disease.

For the images presented, as well as intermediate data collection images of the neighboring tissue regions, we did not observe clear electron microscopy hallmarks of myelin degeneration previously described in mouse models, such as ballooned myelin, abnormal myelin outfoldings, degenerated sheaths, or separate sets of sheaths with a large space between (Gu et al., Brain Research Bulletin, 2018). Based on these observations, we believe that the ultrastructure of myelin largely reflects physiological conditions.

However, to acknowledge the potential influence of Alzheimer's disease pathology on myelin ultrastructural changes, we have added a clarification in the discussion (lines 524-529). While the observed myelin ultrastructure is primarily interpreted as physiological, we acknowledge the possibility that Alzheimer's disease pathology may contribute to some extent to myelin ultrastructural changes in our data. We thank the reviewer for prompting this discussion and

ensuring a nuanced consideration of the myelin observations in the context of Alzheimer's disease neuropathology.

6. The Results text jumps between the use of past and present text. The manuscript would be easier to follow if the results were all written in the past tense, as is standard.

We thank the reviewer for pointing this out. We have revised the results section to be consistently past tense.

7. The sentence referring to the study by Gilbert et al. in BioRxiv beginning line 345, 'That study obtained novel subtomogram averages of in situ tau fibrils, which raise intriguing questions about differences between in situ and ex vivo tau fibril averages,' requires clarification. In that study, the authors were able to fit ex vivo tau fibril structures into all of their low-resolution subtomogram averages, without any noticeable differences.

We thank the reviewer for highlighting this. Extended Fig. 10 of Gilbert et al. provides a subtomogram average that does not fit with any of the presented *ex vivo* tau fibril structures. Additionally, while known paired helical filament structures fit well into each sub-tomogram average of Extended Fig. 10a-e, there are differences between those averages that may indicate subtle differences between subpopulations of paired helical filaments. We have revised the sentence to read "That study obtained several different subtomogram averages of in situ tau fibrils, highlighting the importance of utilizing cryo-ET to investigate tau filament structure and biology in primary tissue context." (lines 532-535).

Reviewer #3 (Remarks to the Author):

The manuscript describes a novel approach to plunge-freeze human tissue samples, and to perform subsequent plasma-FIB milling and cryo-ET. This is one of the first examples where it is shown to be possible to obtain fresh human tissue samples, and vitrify them in an accessible way by plunge-freezing. The authors describe a protocol for how to achieve the vitrification of thicker tissue samples (~100-200 um), and also describe in detail how to create lamellae with the help of plasma FIB milling. This manuscript is definitely interesting for the general audience. The manuscript does show the applicability of this new sample preparation method. It is impressive to see that a biopsy of a human brain can be vitrified by just plunge freezing and turned into lamellae, and then imaged by cryo-ET.

A few minor changes could further improve the manuscript:

- Ext. data figure 3A and Figure 2A do look like there is some crystalline ice (dark shadows). Could you please comment whether those come just from ice contamination on top of the lamellae, or if they are areas of local non-vitreous ice (and how often this was encountered)? To show if this is contamination or not, could you please show those (or similar) example slices alongside with the xz and/or yz slices through the tomogram?

We appreciate the reviewer's keen observation and request. These high-density features are not indicative of surface ice contaminations. To clarify this, we have followed the reviewer's recommendation to include orthogonal xy, yz, and xz slices through the tomogram to demonstrate that these features are embedded inside the lamella. Additionally, to examine whether these high-density features are local non-vitreous ice, we have included tilt-series images to show that these features do not exhibit changing contrast at different tilt angles (Supplementary Fig. 2i,k,m,o), as would be expected for crystalline ice due to Bragg reflections (Supplementary Fig. 2a,c,e,g). Based on these observations, we lean towards the conclusion that the observed high-density features are likely biological materials within the brain tissue.

- The introduction (Lines 39-80) should be more concise. The concepts important to the paper should be introduced in a more succinct way.

We appreciate this comment and have revised the introduction to be more streamlined and focused.

- Please use 3D for Three-dimensional and 2D for Two-dimensional both in the text in figure legends for the sake of space and clarity.

We have made these changes throughout the revised manuscript.

- The methods section says “Tilt series were aligned using platinum deposition on the surface of the lamellae as fiducials and reconstructed into tomograms via weighted back projection in IMOD”. Some of the tomographic reconstructions look like the tilt series alignment was a bit suboptimal (videos 1-2). Could you please use patch tracking (with minimal number of patches 2-4, or even just coarse alignment!), and see if the reconstruction will look better? alignment based on platinum fiducials often skews the center of the tomogram where your features of interest are. If it's better, then please show the new reconstructions.

We thank the reviewer for this helpful suggestion and have indeed explored both coarse alignment only and patch tracking approaches. However, for these datasets, which feature repetitive patterns of myelin cross sections and high-density platinum depositions, the application of biological-feature-driven cross-correlation methods, including patch tracking, has resulted in poorer tracking and alignment outcomes.

- Could you please report on the “statistics”: what was the number of brain samples that you obtained in total, how many lamellae were produced? How much FIB-SEM time did it take approximately? How many lamellae were in the end imaged? And how many successful tomograms were acquired? This is important to know for someone who might like to use your new approach.

We thank the reviewer for this thoughtful request. We have added this information throughout the text of the revised manuscript in the corresponding locations. The number of brain samples is 4 (line 159). The amount of time is listed for both parts of the milling protocol at lines 236-238 and lines 258-259. The number of lamellae imaged by transmission EM with the vitrification protocol is 17 (line 164). The number of lamellae produced and imaged with the final FIB protocol is 9 (line 277). The number of successful reconstructed tomograms for the final FIB protocol is 40 (line 397).

- What was the average lamellae thickness? Please report this in the methods section or in the results section.

We have newly reported the average lamella thickness (350 nm) as measured by tomogram reconstruction in the results section (lines 293-297).

- Could you please show a segmentation of the myelin membranes in one of the figures/videos? This would really improve the visual read out of the paper. Since most observations are descriptions, the 3D segmentation could really aid the reader.

We thank the reviewer for the suggestion. In the revised manuscript, we have included a new movie (Supplementary Movie 5) showing the unannotated tomogram followed by the 3D segmentation of the myelin membranes shown in Fig. 5a.

Reviewers' Comments:

Reviewer #1:

Remarks to the Author:

I am happy with the revision. I find the new version has better focus.

I am happy to recommend publication.

Alex de Marco

Reviewer #2:

Remarks to the Author:

The authors have done an excellent job in addressing all of my comments, with the exception of comment 7, relating to the sub-tomogram averages in (Gilbert et al. BioRxiv 2023).

I maintain that these subtomogram averages are not sufficient to conclude that in situ tau fibrils have different structures to ex vivo tau fibrils. The resolution of the subtomogram averages is extremely low and it is not clear if they actually represent tau, or a filamentous assembly of some other protein.

It is not clear if there are differences between in situ and ex vivo tau fibrils, although this is something that subtomogram averaging may be able to address if the resolution is improved. I suggest that the sentence, "That study obtained several different subtomogram averages of in situ tau fibrils, highlighting the importance of utilizing cryo-ET to investigate tau filament structure and biology in primary tissue context," is altered to reflect these current uncertainties.

Other than this point, I enthusiastically support the publication of this work in its revised form.

Reviewer #3:

Remarks to the Author:

The authors have answered my questions fully, and improved the manuscript

REVIEWER COMMENTS

Reviewer #1 (Remarks to the Author):

I am happy with the revision. I find the new version has better focus.
I am happy to recommend publication.

Alex de Marco

We thank Dr. de Marco for the suggestions and help improving the manuscript.

Reviewer #2 (Remarks to the Author):

The authors have done an excellent job in addressing all of my comments, with the exception of comment 7, relating to the sub-tomogram averages in (Gilbert et al. BioRxiv 2023).

I maintain that these subtomogram averages are not sufficient to conclude that in situ tau fibrils have different structures to ex vivo tau fibrils. The resolution of the subtomogram averages is extremely low and it is not clear if they actually represent tau, or a filamentous assembly of some other protein.

It is not clear if there are differences between in situ and ex vivo tau fibrils, although this is something that subtomogram averaging may be able to address if the resolution is improved. I suggest that the sentence, "That study obtained several different subtomogram averages of in situ tau fibrils, highlighting the importance of utilizing cryo-ET to investigate tau filament structure and biology in primary tissue context," is altered to reflect these current uncertainties.

We thank the reviewer for emphasizing this point. For clarity and simplicity, we have decided to remove this sentence and refocus the paragraph to be a general discussion of the advantages of developing complementary approaches for studying primary human tissue via cryo-EM.

Other than this point, I enthusiastically support the publication of this work in its revised form.

Reviewer #3 (Remarks to the Author):

The authors have answered my questions fully, and improved the manuscript

We thank the reviewer for their suggestions to improve the manuscript.